# Phase II multicentre, double-blind, randomised trial of ustekinumab in adolescents with new-onset type 1 diabetes (USTEK1D): trial protocol

John W Gregory [ORCID],[1] Kymberley Carter [ORCID],[2] Wai Yee Cheung [ORCID],[3] Gail Holland [ORCID],[2] Jane Bowen-Morris,[4] Stephen Luzio [ORCID],[3] Gareth Dunseath [ORCID],[3] Timothy Tree [ORCID],[5,6] Jennie Hsiu Mien Yang [ORCID],[5,6] Ashish Marwaha [ORCID],[7] Mohammad Alhadj Ali,[4] Nadim Bashir [ORCID],[2] Hayley Anne Hutchings [ORCID],[2] Greg W Fegan [ORCID],[2] Rachel Stenson,[4] Stephen Hiles [ORCID],[2] Susie Marques-Jones,[8] Amy Brown,[9] Danijela Tatovic,[4] Colin Dayan [ORCID] [4]

**Correspondence to**
Dr Kymberley Carter;
k.carter@swansea.ac.uk

## ABSTRACT

**Introduction** Most individuals newly diagnosed with type 1 diabetes (T1D) have 10%–20% of beta-cell function remaining at the time of diagnosis. Preservation of residual beta-cell function at diagnosis may improve glycaemic control and reduce longer-term complications. Immunotherapy has the potential to preserve endogenous beta-cell function and thereby improve metabolic control even in poorly compliant individuals. We propose to test ustekinumab (STELARA), a targeted and well-tolerated therapy that may halt T-cell and cytokine-mediated destruction of beta-cells in the pancreas at the time of diagnosis.

**Methods and analysis** This is a double-blind phase II study to assess the safety and efficacy of ustekinumab in 72 children and adolescents aged 12–18 with new-onset T1D.

Participants should have evidence of residual functioning beta-cells (serum C-peptide level >0.2nmol/L in the mixed-meal tolerance test (MMTT) and be positive for at least one islet autoantibody (GAD, IA-2, ZnT8) to be eligible.

Participants will be given ustekinumab/placebo subcutaneously at weeks 0, 4 and 12, 20, 28, 36 and 44 in a dose depending on the body weight and will be followed for 12 months after dose 1.

MMTTs will be used to measure the efficacy of ustekinumab for preserving C-peptide area under the curve at week 52 compared with placebo. Secondary objectives include further investigations into the efficacy and safety of ustekinumab, patient and parent questionnaires, alternative methods for measuring insulin production and exploratory mechanistic work.

**Ethics and dissemination** This trial received research ethics approval from the Wales Research Ethics Committee 3 in September 2018 and began recruiting in December 2018.

The results will be disseminated using highly accessed, peer-reviewed medical journals and presented at conferences.

**Trial registration number** ISRCTN14274380.

## Strengths and limitations of this study

► This trial is being undertaken in 16 sites across the UK (England=12, Scotland=2 and Wales=2) with a recruitment period of 3 years due to temporary closure of recruiting sites caused by COVID-19 infection.

► The trial will provide evidence of the efficacy and safety of treating new-onset type 1 diabetes in 12–18 years with ustekinumab (STELARA).

► We have included an extensive range of secondary and exploratory outcomes to investigate the efficacy and safety of ustekinumab, and to expand existing methodological designs in this field.

## INTRODUCTION

Nearly 100 years after the discovery of insulin, over 70% of patients with type 1 diabetes (T1D) continue to have unsatisfactory glycaemic control putting them at risk of long-term complications.[1] Tragically, death rates among adolescents have not improved in the last few decades (1968–2009).[2] Despite major advances in closed loop insulin pump therapy, much of the morbidity arises from young people failing to engage with complex therapies.

Most individuals have 10%–20% of beta-cell function remaining at the time of diagnosis of T1D.[3] Preservation of even 5% of beta-cell function has been shown to lower blood glucose levels (as measured by HbA1c tests) by 1%, permit over 50% of people to reach target glycaemic levels, reduce hypoglycaemic risk by >50% and reduce long-term complications by 50%.[4] [5] Immunotherapy has the potential to preserve endogenous

beta-cell function and thereby improve metabolic control even in poorly compliant individuals.[6–8]

Novel low-risk targeted biological therapies are widely used in other autoimmune diseases such as rheumatoid arthritis, psoriasis and inflammatory bowel disease, but no treatment was licensed for use in T1D. Ustekinumab is licensed in the UK for the treatment of psoriasis in children and adults, psoriatic arthritis in adults and Crohn's disease in adults.

Extensive evidence exists to implicate two major autoimmune cytokine pathways, IL-12/IFN-γ and IL-23/IL-17, in beta cell destruction. Ustekinumab (STELARA), binds and inhibits the p40 molecular subunits of both IL-12 and IL-23 thus blocking their action in inducing pathogenic CD4 Th1 and Th17 T cell subsets.[9] Our overarching hypothesis is that interrupting the IL-17 and IFN-γ axes in individuals with recent-onset T1D will halt or slow the autoimmune destruction of beta cells sufficiently to permit beta cell preservation and maintain residual physiological insulin secretion. Given the therapeutic success of biologics that target immune molecules in other autoimmune and inflammatory diseases, and the evidence that IL-17 and IFN-γ producing cells are pathogenic to beta cells, we propose that ustekinumab may be beneficial for the treatment of T1D.

This paper presents the protocol for a double blind, multicentre, randomised phase II trial to evaluate the effect of ustekinumab in patients aged 12–18 years with new-onset T1D.

## METHODS
### Overview
This is a multicentre, double blind, randomised, controlled trial comparing ustekinumab with placebo (2:1 ratio). Doses of ustekinumab will be 2 mg/Kg body weight if the child is ≤40 Kg and 90 mg if >40 Kg. Doses will be administered at week 0, 4, 12, 20, 28, 36 and 44 and with follow-up at week 52 (see figure 1). The study will be carried out in 12–18 year olds within 100 days of diagnosis of T1D in 16 sites across mainland UK.

The primary objective is to determine the efficacy of ustekinumab for preserving mixed-meal tolerance test (MMTT) stimulated 2-hour plasma C-peptide area under the curve (AUC) at week 52 as compared with placebo. This follows the rationale published by Greenbaum et al.[10]

### Objectives
Table 1 details the trial objectives, outcome measures and time points for data analysis.

### Consent
Potential participants identified from health records, clinical contacts, patient registries and self-referrals through the T1DUK consortium and ADDRESS-2 website will be asked to view our short recruitment video (https://www.youtube.com/watch?v=8kuCefuBSW4), followed by a more detailed information sheet relevant to their age (see online supplemental appendices 1–4).

Written informed consent will be obtained for all participants at the first screening visit (see online supplemental appendices 5–8). For participants under 16 years, written assent will be obtained in addition to written consent from a parent/carer. Reconsent will be requested when participants turn 16 years.

### Eligibility criteria
Consented participants will have eligibility checks (see table 2), including autoantibody screening and an MMTT. Tuberculosis (TB) must be ruled out using a chest X-ray and either a Mantoux test or a blood-based TB test. All blood and urine tests must be within clinically normal parameters.

The first dose of investigational medical product (IMP) must be given within 100 days from clinical diagnosis. The screening MMTT must be within 37 days of the first dose of IMP.

### Randomisation and blinding
Minimisation by age (12–15 vs 16–18 years, respectively) and screened peak C-peptide levels (0.2–0.7 vs >0.7 nmol/L) will be used to ensure balance between treatment groups. These variables are important prognostic factors and need to be evenly distributed between the groups. The baseline C-peptide cut-off of 0.7 nmol/L was selected to correspond with Lachin et al.[11]

The treatment:placebo ratio will be 2:1 to promote recruitment and to provide additional data on drug safety. The minimisation algorithm and randomisation list will be provided by Sealed Envelope (https://sealedenvelope.com) and accessed by sites using an online randomisation system which was validated prior to use by statisticians in Swansea Trials Unit (STU). The system will email a randomisation code to designated site personnel including pharmacy who will cross-reference it with a code break list to determine the allocation.

Dosage and regimen of placebo and ustekinumab will be matched. Only staff preparing the blinded syringe will be unblinded at sites. Participants, research staff and the trial office remain blinded, with only limited independent researchers at STU managing the code break list and any IMP-related queries from pharmacies.

Emergency unblinding will be managed by Sealed Envelope. If emergency unblinding is delayed, the treating clinician should treat the patient as if ustekinumab has been given.

### Trial assessments
An overview of the trial procedures are listed in table 3.

Further details are provided below for each assessment contributing towards the objectives.

#### Mixed-meal tolerance test
Secretion of C-peptide will be assessed for the primary outcome measure of the trial using an MMTT at screening and week 52. We also conduct an MMTT at week 28 to address a secondary objective.

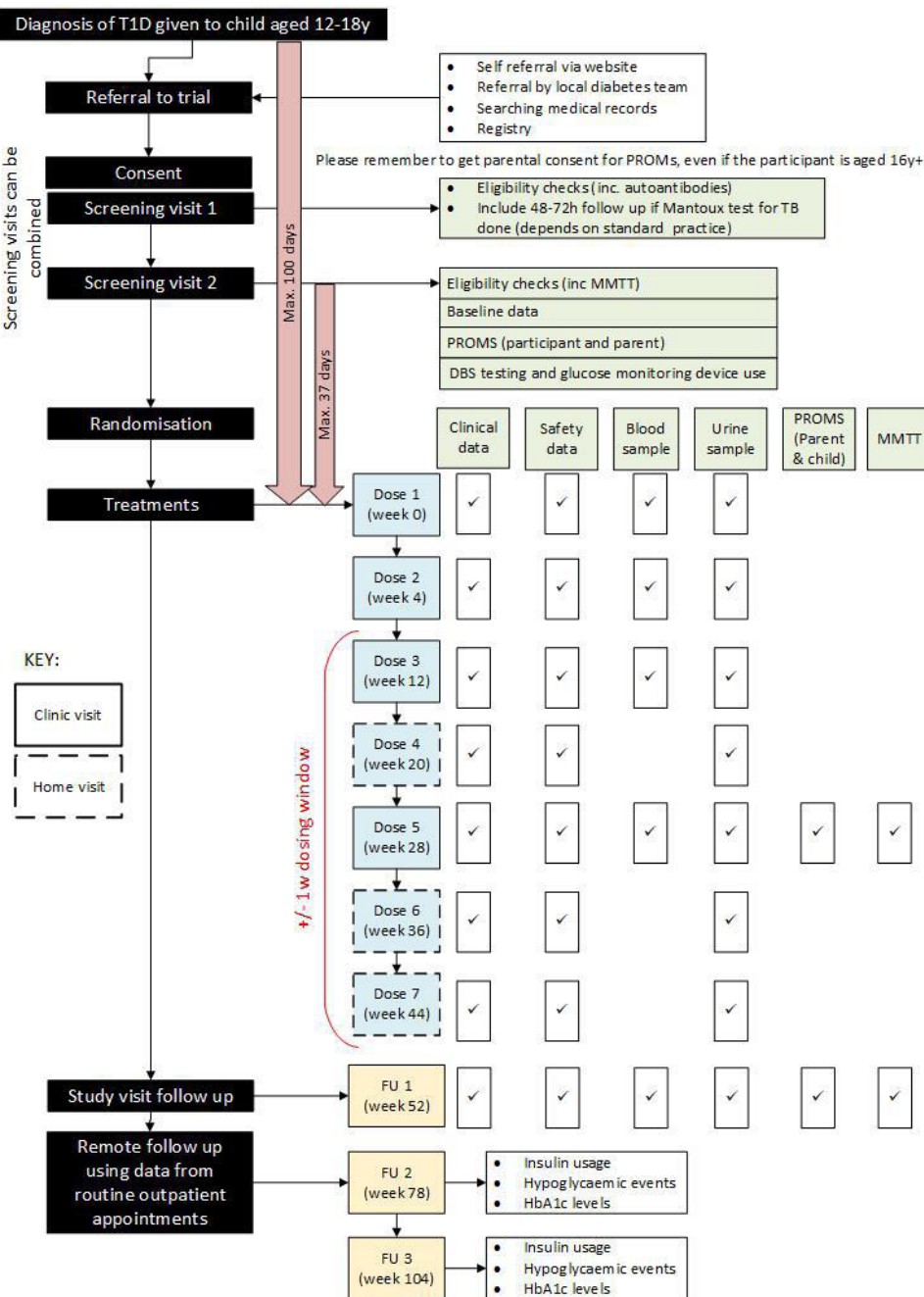

**Figure 1** Trial flow chart. DBS, dried blood spot; MMTT, mixed-meal tolerance test; PROM, patient-reported outcome measure; T1D, type 1 diabetes; HbA1c, glycosylated haemoglobin.

Participants will fast from midnight and check their blood glucose on waking. The MMTT will be started between 7:00 and 11:00 hours if their blood glucose prior to arriving is between 4.0 and 11.1 mmol/L (inclusive). If it is <4 mmol/L on waking, the test will be postponed to a different day. If the value is >11.1 mmol/L the participant will be advised to take an appropriate correction bolus of very short acting insulin at home so that the blood glucose would be within range on arrival at the hospital. The test may be postponed if the blood glucose is not in range after 2 hours.

The participant will be asked to drink a standardised liquid meal provided by the trial—Ensure Plus 6 mL/kg (Maximum 360 mL). This must be ingested within 5 min.

Blood glucose measurements will be taken prior to, and at the end of, the MMTT. The participant will void their bladder and urine will be collected at the end of the MMTT (at 120 min). Venous blood samples will be collected for measurement of C-peptide at time 0, 15, 30, 60, 90 and 120 min. Blood samples for mechanistic work will be taken at time 0.

## Glucose monitoring

All participants will be provided with an Abbott FreeStyle Libre blood glucose monitoring system. Participants are expected to wear a sensor for at least 2 weeks prior to each study visit and will be advised to read their measurements

**Table 1** Objectives and outcome measures

| Objectives | Outcome measures |
| --- | --- |
| **Primary objective** | |
| To determine the efficacy of ustekinumab (dose: 2 mg/kg (≤40 kg); 90 mg (>40 kg)) for preserving MMTT stimulated 2 hour C-peptide area under the curve (AUC) at week 52 as compared with placebo in children and adolescents with new-onset T1D. | MMTT C-peptide AUC values at week 52 |
| **Secondary objectives** | |
| 1. To determine the efficacy of the ustekinumab dosing to elicit response to treatment. | No of responders (defined as participant who has glycosylated haemoglobin (HbA1c) ≤48 mmol/mol and mean daily insulin use <0.5 IU/kg/day) measured over seven consecutive days during the 2 weeks preceding the visit in treatment and placebo group at week 52 |
| 2. To investigate additional efficacy (metabolic) endpoints including MMTT C-peptide AUC at week 28, HbA1c and insulin use measurements at week 52. | MMTT C-peptide AUC values at Week 28 |
| | HbA1c at weeks 0, 12, 28 and 52 |
| | Exogenous insulin requirement as reflected in mean daily insulin usage over seven consecutive days (IU units/kg body weight/day) as recorded in diaries prior to study visits at weeks 12, 28 and 52 |
| | Insulin dose adjusted HbA1c (IDAAC) at week 52 |
| 3. To compare alternative metabolic endpoint assays to MMTT: including glycaemic variability in glucose monitoring systems (Freestyle Libre) and hypoglycaemia rates. | Glycaemic variability parameters downloaded from glucose monitoring at each study visit, for example, ► Blood glucose level at 1,2,3 hours before and after each meal. ► No of episodes and length of time within the following glucose level: below 4.0 mmol/L,>10 mmol/L and >15 mmol/L. ► % Time hypoglycaemic (<3.0 mmol/ and <4.0 mmol). |
| | Clinical hypoglycaemic events determined by patient diary reports and AE reports at week 52 |
| 4. To determine the safety of ustekinumab dose in adolescents with new-onset T1D. | Frequency and severity of all adverse events at week 52 |
| 5. To compare between treatment arms and across the course of treatment the age appropriate PROMs scores completed by participants and parents/carers. | HYPOFEAR, DTSQ and PedsQL questionnaires completed by participants and their parent/carer at weeks −2, 28 and 52 |
| **Exploratory objectives** | |
| 1. To investigate alternative ways of measuring insulin production other than MMTT C-peptide. | Proinsulin, glucagon, somatostatin levels, dried blood spot (DBS) C-peptide at weeks 28 and 52 |
| | DBS C-peptide vs MMTT C-peptide at weeks −2, 28 and 52 |
| 2. To investigate changes in relevant immune mechanistic parameters include flow cytometry immune phenotyping of all Interleukin (IL)-17 and Interferon-gamma (IFN-g) secreting T cell subsets, fluorospot analysis for IL-17 and IFN-g secretion in response to antigens for CD4 +T cells. | Changes at weeks 12, 28 and 52 in: ► Immune phenotype of all IL-17, IFN-g secreting immune subsets. ► T cell responses to antigens or peptides derived from islet antigens (including proinsulin, Glutamic Acid Decarboxylase (GAD) and Islet antigen 2 (IA-2) measured by cytokine FLUOROSPOT (IFN-g and IL-17). ► T cell responses to antigens or peptides derived from islet antigens (including proinsulin, GAD, IA-2) measured by the level of IFN-g, IL-17, IL-12 and IL-23 production in supernatants (Luminex). ► Additional immunological biomarkers (eg, flow cytometry profiles, T cell responsiveness measured by activation profiles, T reg assays, autoantibodies). |
| 3. To investigate ustekinumab pharmacokinetics (PK) and compliance with therapy | Ustekinumab drug levels in serum at weeks 4, 12, 28 and 52 |
| 4. To explore association of C-peptide changes with age-appropriate PROMs. | C-peptide AUC and HYPOFEAR, DTSQ and PedsQL questionnaires at weeks −2, 28 and 52 |
| 5. To compare participant and parent/carer proxy completed PROMs. | HYPOFEAR, DTSQ and PedsQL questionnaires completed by participants and their parent/carer at weeks −2, 28 and 52 |
| 6. To investigate the longer term effect of ustekinumab on glycaemic control. | ► Severe hypoglycaemic events. ► Insulin use. ► HbA1c. ► C-peptide using DBS samples. ► Continuous Glucose Monitoring (CGM) data. At weeks 78 and 104 |

Table key: −2 refers to the second screening visit about 2 weeks prior to dose 1.
AE, adverse event; DTSQ, Diabetes Treatment Satisfaction Questionnaire; HYPOFEAR, Hypoglycaemia Fear Survey; MMTT, mixed-meal tolerance test; PedsQL, Paediatric Quality of Life inventory; T1D, type 1 diabetes.

**Table 2** Trial eligibility criteria

| Inclusion criteria | Exclusion criteria |
|---|---|
| Clinical diagnosis of immune-mediated type 1 diabetes (T1D) mellitus as defined by the American Diabetes Association (ADA).[25 26] | Breastfeeding, pregnancy or unwillingness to comply with contraceptive advice and regular pregnancy testing throughout the trial. |
| Commenced on insulin within 1 month of clinical diagnosis (defined as confirmed raised blood sugar (ADA criteria), not symptoms alone). | Prior exposure to ustekinumab within 3 months of the first dose of Investigational Medical Product (IMP). |
| An interval of ≤100 days between the confirmed diagnosis (defined as date of first insulin dose) and the first planned dose of the IMP. | Use of more than 10 mg prednisolone daily (or equivalent) for >5 days within 3 months of the first dose of IMP. |
| Written and witnessed informed consent/assent to participate. | Prior exposure to any anti-lymphocyte monoclonal antibody, such as anti-CD20, antithymocyte globulin, rituximab (Rituxan) or alemtuzumab (Campath). |
| Male or female, aged 12–18 years inclusive at the time of randomisation. | Use of immunosuppressive or immunomodulatory therapies, including systemic steroids within 30 days prior to receiving the first dose and/or intent on using any monoclonal antibody therapy given for any indication for the duration (including follow-up) of the trial. |
| Evidence of residual functioning beta-cells (peak serum C-peptide level >0.2 nmol/L in the MMTT test). | Use of any hypoglycaemia agents other than insulin, for more than 6 weeks, at any time prior to trial entry, including SGLT2 inhibitors. |
| Positive for at least one islet autoantibody (GAD, IA-2, zinc transporter protein 8 (ZnT8). | Use of inhaled insulin. |
| Body weight <100 kg. | Known alcohol abuse, drug abuse. |
| Willing to record all insulin doses and blood glucose levels required for monitoring during the study, including reporting any hypoglycaemic events. | Evidence of active Hepatitis B, Hepatitis C, HIV or considered by the investigator to be at high risk for HIV infection. |
| Willing to provide dried blood spot samples. | Significant systemic infection during the 6 weeks before the first dose of the IMP (eg, infection requiring hospitalisation, major surgery, requiring IV antibiotic treatment). Other infections, for example, glandular fever, bronchitis, sinusitis, cellulitis or urinary tract infections must be assessed on a case-by-case basis by the investigator to assess whether they are serious enough to warrant exclusion or delay to inclusion. |
| Willing to wear the FreeStyle Libre Glucose Monitor device at least 2 weeks prior to a study visit. | History of current or past active tuberculosis (TB) infection and no latent TB. Active TB will be assessed using a mandatory chest X-ray and one of the following: a) blood-based test; b) the Mantoux skin test. |
| Willing to complete a diary and quality of life questionnaires. | Any live immunisations for 1 month prior to trial entry. Planned live immunisations are also not permitted during the study period. |
| Willing to consent to remote follow-up via health records and telephone contact. | Previous use of any other investigational drug within the 3 months prior to the first dose and/or intent on using any investigational drug for the duration (including follow-up) of the trial. |
| Female participants have a negative urine test for pregnancy; all participants must agree to use adequate contraception if they become/are sexually active (hormonal-based contraception, double barrier contraception, abstinence) until 4 months following the date of their final treatment of IMP. | Recent (within 3 months) involvement in other research studies, which, in the opinion of investigators, may adversely affect the safety of the participants or the results of the study. |
| | Significantly abnormal laboratory results during the screening period, other than those due to T1D. |
| | Prior allergic reaction, including anaphylaxis, to any component of the IMP product. |
| | Prior allergic reaction, including anaphylaxis, to any human, humanised, chimeric or rodent antibody treatment. |
| | Any major planned surgery scheduled within the 30 day period prior to the first drug dose or anticipating requiring major surgery during the study period. |
| | Any other medical condition or treatment that, in the opinion of investigators, could affect the safety of the participant's participation or outcomes of the study, including malignancy, immunocompromised states and autoimmune conditions. |
| | Participants or parents/carers who lack the capacity to comply with trial requirements. |

IMP, investigational medical product; MMTT, mixed-meal tolerance test.

at least 4–7 times a day. Anonymised data will be sent electronically to the trial office.

## Hypoglycaemia

Participants will be advised by the research staff to record in a trial diary any hypoglycaemia symptoms between each study visit. This will be compared with glucose monitoring data. Participants will be asked to record a finger-prick blood glucose in the diary any time hypoglycaemic symptoms occur, even if the glucose monitor sensor is also being worn. A medic will categorise all hypoglycaemic events recorded in the diary according to American Diabetes Association (ADA) guidelines.[12 13]

## Dried blood spot measurements

Dried blood spot (DBS) sampling will be carried out at home by the participant weekly from screening until week 28 and then monthly up to month 12 for the measurement of C-peptide. Blood samples will be obtained by finger prick and placed onto filter paper cards (Perkin

**Table 3** Schedule of events at sites

| | Screening (SC)* | | Dose | | | | | | | | Follow-up |
|---|---|---|---|---|---|---|---|---|---|---|---|
| | SC1 | SC2 | 1 | 2 | 3 | 4 | 5 | 6 | 7 | | |
| Visit | SC1 | SC2 | 1 | 2 | 3 | 4 | 4 | 6 | 7 | 8 | |
| Week | Approx. −2 | | 0 | 4 | 12 | 20 | 28 | 36 | 44 | 52 | |
| Window allowed | | | ≤100 days of clinical diagnosis and ≤37 days of SC2 | ±1 week | | | | | | | |
| Consent | X | | | | | | | | | | |
| Medical History | X | | | | | | | | | | |
| Physical exam | X | | X | | X | | X | | | X | |
| Concomitant medication (D) | X | X | X | X | X | X | X | X | X | X | |
| Weight | X | X | X | | X | | X | | | X | |
| Height | X | | X | | X | | X | | | X | |
| Vital signs | X | X | X | | X | | X | | | X | |
| TB tests† | X | | | | | | | | | | |
| Adverse events (including hypoglycaemia) (D) | | X | X | X | X | X | X | X | X | X | |
| Blood draw‡ § | X | X | X | X | X | | X | | | X | |
| Urine collection¶ | X | X | X | X | X | X | X | X | X | X | |
| Dried blood spot review | | X | X | X | X | X | X | X | X | X | |
| Download blood glucose monitoring data | | | (X) | X | X | (X) | X | (X) | X | X | |
| Glycaemic control (as part of routine care) | | | X | X | X | X | X | X | X | X | |
| Insulin dose usage (D) | | X | X | X | X | X | X | X | X | X | |
| PROMs (adolescent and parent) | | X | | | | | X | | | X | |

Table key: (X)=optional data download; (D)=data from diary.
*Screening visits may be combined.
†Chest X-ray and either a (1) blood test (T spot/quantiferon) or (2) Mantoux test.
‡Safety bloods-full blood count; urea, electrolytes and creatinine; liver function tests (total bilirubin, total protein, albumin, aspartate aminotransferase (AST), serum glutamic oxaloacetic transaminase (SGOT), serum glutamic pyruvic transaminase (SGPT), alanine transaminase (ALT), alkaline phosphatase; thyroid stimulating hormone; immunoglobulins (G, A, M); calcium; magnesium, phosphate, lipid profile (total cholesterol, low density lipoprotein (LDL), high density lipoprotein (HDL), triglyceride), HbA1c. For screening only, we also request HIV and Hepatitis B and C and TB testing.
§Bloods for research laboratories: Diabetes Research Unit Cymru (DRUC)=MMTT, Islet autoantibodies, glycosylated haemoglobin (HbA1c), exocrine enzymes, proinsulin; Kings College London=T cell assays, Flow-cytometry profiles of leucocyte populations, cytokine production by CD4 and CD8 T cells; Royal Devon & Exeter Hospital=glucagon and somatostatin levels; University of Bristol=cell free DNA; commercial company=pharmacokinetics analysis.
¶Urine samples are collected for pregnancy testing (females), urinalysis for pH, blood and protein by dipstick urinalysis and laboratory analysis for albumin/creatinine ratio. We also collect a sample as part of the MMTT for DRUC.
MMTT, mixed-meal tolerance test; PROM, patient-reported outcome measure; TB, tuberculosis.

Elmer). Samples will be provided before the first meal of the day, and one 60 min afterwards. Patients will be asked to withhold their pre-meal insulin until after the second DBS samples have been taken.

### Insulin dose
Mean daily insulin use will be calculated over seven consecutive days during the 2 weeks preceding all visits and participants will be asked to record all insulin usage in their diary during those 2 weeks. This value will be calculated in units of IU/kg/day. Where data from consecutive days are not available, the 3 days closest together will be used.

### Body weight and body mass index (clinical care measurement)
Body weight and height will be recorded at site visits and the most recent weight recorded will be used to calculate drug dosages for forthcoming treatment visits. Body mass index will be calculated as standard: weight (kg)/(height (m))$^2$.

## Patient and parent-reported outcome measures

Quality of life for participants and their parent/carer will be assessed at screening, and weeks 28 and 52 by validated questionnaires: the Hypoglycaemia Fear Survey[14 15]; Diabetes Treatment Satisfaction Questionnaire for inpatients[16]; Paediatric Quality of Life inventory Copyright 1998 JW Varni, PhD (generic core scale[17 18] and diabetes-specific[19 20] modules).

The questionnaires will be completed during the latter stages of the MMTT while the participant and parent are waiting for the end of the test. Participant and parent will be encouraged not to discuss their responses with each other.

## Glycaemic control

Glycaemic control will be maintained according to clinical guidelines with the support of the participant's local diabetes clinical care team. HbA1c will be measured as per the study schedule based on the local laboratory results with a target value set according to 2015 National Institute for Health and Care Excellence (NICE) guidelines[21] in agreement with the participant and their clinical care team. Where this target is not met, advice will be given as clinically required.

## Urine C-peptide/creatinine ratio

Urine C-peptide/creatinine ratio will be measured from the 120 min urine sample taken during the MMTT at screening, weeks 28 and 52. We selected this to determine whether it could be used as an alternative non-invasive test for future trials based on successes in other trials.[22 23]

## HbA1c

HbA1c will be tested in the local NHS laboratories of the study sites to guide clinical care. A blood sample will also be taken at weeks 0, 12, 28 and 52 for measurement of HbA1c using an HPLC method.

## Immunological changes (mechanistic study)

Changes in immune mechanistic parameters including IL-17 and IFN-gamma production, phenotypes and function of CD4 +and CD8+T cells will be assessed by flow cytometry immunophenotyping, Fluorospot and other immune assays, such as Luminex, at screening or week 0 (as baseline), and week 12, 28 and 52, using primarily overnight blood samples and also cryopreserved peripheral blood mononuclear cells.

Changes in IL-17 and IFN-gamma production will be measured in both agnostic and antigen-specific manner, where for the latter T cell responses will be determined in response to antigens or peptides derived from islet antigens.

## Long-term follow-up assessments

We will record weight and height, insulin doses over a 2-week period, severe hypoglycaemia events and HbA1c levels at time points closest to weeks 78 and 104 which also coincide with a routine clinic visit. The data will be sourced from the medical records where possible and from the participant using a short questionnaire.

We also seek consent to have two additional DBS cards completed at the corresponding time points and an anonymised copy of the glucose monitor data for the 2 weeks prior to the time points matching the clinic visits.

## TRIAL TREATMENTS
### Ustekinumab (STELARA)

Ustekinumab is a fully human IgG1k monoclonal antibody supplied by the marketing authorisation holder Janssen-Cilag (EU/1/08/494/002). It is supplied as sterile single use 2 mL glass vials containing 0.5 mL of solution with 45 mg of ustekinumab for injection. Section 4.8 of the Summary of Product Characteristics (SmPC) for STELARA (https://www.medicines.org.uk/emc/product/4413/smpc) dated 22 March 2018 will be used as the reference safety information for pharmacovigilance purposes. It was assessed by the Medicines and Healthcare products Regulatory Agency (MHRA) as part of the original approvals process.

The SmPC has been updated three times so far. However, there were no significant change to the safety parameters of the trial so the original version continues to be used.

### Placebo

Saline in the form of sodium chloride 0.9% w/v solution for injection will be used as the placebo. Any brand of saline with a marketing authorisation in the UK can be used for this trial. A representative SmPC will be used to represent all saline (marketing authorisation number PL 02848/0157).

### Discontinuation/modification of drug dosing

Drug dosing will only be altered in response to a change in body weight as per the protocol which states 2 mg/Kg≤40 Kg or 90 mg if >40 Kg.

## WITHDRAWALS

The principal investigator (PI) or participant (or parent/carer if the participant is <16 years) can opt to discontinue treatment for any reason. The participant (and parent if <16 years) will be asked to remain in the trial for sample and data collection purposes only. They have the right to withdraw completely without giving a reason.

Oversight committees and the sponsor can request the withdrawal of a participant(s) or to terminate the trial.

Exceeding the time frame for receiving medication may also result in withdrawal from treatment.

## SAFETY REPORTING

The risk of major adverse unexpected events is anticipated to be low. Ustekinumab has a marketing authorisation in the age group being studied for other indications. The available SmPC describes all essential information for the use

of the medicine, and the qualitative and quantitative information on benefits and risks. Participants being exposed to ustekinumab are a different disease population from those described in the SmPC. In addition, the dose used in this trial is higher than that currently licensed for psoriasis in adolescents, although it (and higher doses) have been used in adults with both psoriasis and Crohn's disease.

Hypoglycaemic events are common in this population and may not necessarily be IMP related. Hypoglycaemia rates are an important secondary outcome, as it is anticipated that these should be reduced by ustekinumab if it is effective. Hypoglycaemic events are recorded specifically for this trial separately from other adverse events (AEs) because they require medical assessment according to ADA guidelines.[12 13]

A review of AEs will be performed at all visits (participant-reported) and using blood and urine samples at screening and 0, 12, 28 and 52 weeks. A urine pregnancy test will be completed on all females at all trial visits. PIs will be expected to assess any values outside the laboratory reference range for clinical significance.

Hypoglycaemia and diabetic ketoacidosis are considered expected for newly diagnosed patients with T1D. If the event leads to death, this will be considered unexpected.

Any pregnancies for female participants or the pregnant partners of male participants must be reported immediately. Pregnant participants will be withdrawn from treatment and asked to provide consent to follow up the pregnancy until the child is 12 months old.

## POST-TRIAL CARE

Following completion of their trial participation, participants will be kept informed of ongoing trial developments including final outcomes following statistical analyses. Should participants be concerned about implications arising from their trial participation, they will be asked to discuss these with their local clinicians. Senior members of the trial team will be available for further advice should the local clinician require.

Once the trial is complete (defined as last participant, completing the 24 months follow-up data collection task), following unblinding, individual participants and their local clinicians will be informed by letter on request as to which arm of the trial they were randomised to. After completing the first 52 weeks of the trial, clinical care and follow-up will be provided by the participant's local diabetes care team. Ustekinumab will not be available for ongoing therapy.

## STATISTICS AND DATA ANALYSIS
### Sample size considerations

The power calculation closely follows Lachin et al[11] based on data for children and young adolescents aged 13–17 years as well as the T1DAL study in 12–35 years.[24] A sample size of 66 apportioned in a 2:1 ratio has a greater than 85% power to detect a 0.2 nmol/L difference between the 2-hour MMTT mean AUC C-peptide values of the

intervention and placebo arms which are assumed to be 0.5 and 0.3 (nmol/L), respectively, at 12 months. Seventy-two participants (48 ustekinumab:24 placebo) will be recruited to allow for approximately 10% lost to follow-up.

## Data analysis

Data cleaning and preparation processes will be carried out prior to final analysis. A statistical analysis plan approved by the data safety monitoring board (DSMB) will be followed.

All participants enrolled will be followed up and included unless they withdraw from the study before the administration of the first dose. An intention-to-treat (ITT) analysis will be carried out. Per-protocol analysis of the primary outcome will also be carried out alongside the ITT analysis if deemed necessary by the trial steering committee (TSC).

The primary data analysis will be the application of analysis of covariance to the 12-month recorded AUC mean values of C-peptide taking into account the baseline values of these measures and using transformations as suggested by Lachin et al.[11] The analysis will be adjusted by important covariates such as gender, age at recruitment, baseline insulin use and glycaemic control.

For the secondary outcomes including the mechanistic and questionnaire studies we will evaluate the various outcomes using the most appropriate statistical approach that is, binomial or logistic regression for binary outcomes, Poisson or related count outcome models for number of events/objects and linear models for continuous outcomes. Where necessary, mixed or multilevel models will be used to account for correlation within observations.

No interim analysis is planned. No subgroup analysis is planned. Should there be substantial non-fidelity to allocated treatment, a per-protocol analysis for the primary outcome will be considered after approval by the TSC.

Efficacy analyses will be adjusted by gender, age and baseline test values. Safety analysis will not be adjusted.

Interim analysis on safety data only will be conducted if requested by TSC/DSMB. Decision criteria based on safety as part of a guideline for early stopping or other adaptations will be set by TSC with input from DSMB.

Every attempt will be made to minimise missing data, encouraging participants to provide week 52 data even if they are no longer taking the interventional medication. Patterns and level of missing data will be examined. Multiple imputation will be considered if required, if there are more than 5% and less than 10% (>3 and <7 participant) missing.

## DATA MANAGEMENT

Source documents produced for this trial will be filed with the participant's medical records. Source data will be entered into trial-specific database of electronic case report forms (eCRFs) at the end of each trial visit within a site agreed timespan. These eCRFs will be coded with the participants study number and will not include patients' names and addresses and will conform to general data protection requirements (GDPR). This database (MACRO V.4.7 Elsevier, 2017) will be hosted on a Swansea University

server with back up and restoration procedures in place. All paper CRFs can be found by logging into the trial website and entering the password.

The trial database will be managed and operated as required by Good Clinical Practice (GCP). The site investigator or delegate will record all study data using the trial specific electronic database provided by STU. All data will be handled and stored in accordance with GDPR, Data Protection Act and applicable legislation.

Data will be checked according to the trial Data Management Plan and queries will be generated and sent to the site investigator for response using the database.

Data from laboratories and the anonymised glucose monitoring and diary data from patients will be securely transferred to the trial office.

Remote data collection after week 52 will be done using the REDCap database with links to participant questionnaires emailed by site researchers. No identifiable data are collected in the database during remote follow-up.

The Chief Investigator and trial statistician will have access to the final dataset for analysis. Should Principal Investigators or others require access to the final dataset this will require approval by the Trial Management Group (TMG), TSC and sponsor.

The trial data will be held in a data repository, the location of which is still being negotiated.

## MONITORING

Monitoring of this trial to ensure compliance with GCP and scientific integrity will be conducted by STU via central and on-site monitoring as per the Trial Monitoring Plan.

This will include 100% central monitoring of all primary outcome data, with site initiation and closedown visits for all sites, and a minimum of one monitoring visit during the recruitment period to complete 100% source data verification on primary outcome data. In addition, the trial office will facilitate monitoring by local Research and Development (R&D) departments at any of the trial sites, should this be requested.

## DISSEMINATION

A publication plan will be developed to organise the outputs from this trial. Outputs will be disseminated using highly accessed, peer-reviewed medical journals and will be presented at conferences.

Authorship will be agreed on by the CI, PIs and members of the TMG and will follow the guidance provided by the International Committee of Medical Journal Editors.

## PATIENT AND PUBLIC INVOLVEMENT

We recruited a panel of children with T1D to help us develop a short recruitment video.

We recruited six patient and public involvement (PPI) representatives, two for each committee (TMG, DSMB and TSC). All PPI representatives are either parents of children with T1D or have T1D themselves. Our PPI representatives assist in the development of participant facing documentation, support applications for approvals and will review and help to disseminate our results.

## STUDY MANAGEMENT

The trial office is based at STU, with the chief investigator, paediatric lead and adult lead all working at Cardiff University.

The sponsor of the trial is Cardiff University. The sponsor can be contacted at resgov@cardiff.ac.uk.

The sponsor has arranged appropriate insurance and indemnity to meet the potential legal liability for harm to the participants arising from the design or management of the trial for negligent harm. In addition, the trial health professionals hold substantive or honorary National Health Service (NHS) contracts, giving them the protection of the appropriate NHS clinical negligence arrangements.

## TRIAL COMMITTEES

The trial oversight committees are the DSMB and TSC who will meet biannually. They comprise of clinical experts, a statistician and public and patient involvement (PPI) representatives and each work to a preagreed charter. The DSMB to provide ethical and safety reviews (including the assessment of AEs and protocol deviations) and the TSC will have general oversight of the trial to ensure recruitment, treatment and follow-up visits are safe and providing the relevant data, and that the protocol is being adhered to, based on DSMB recommendations.

The TMG consists of the trial team, independent advisors and PPI representatives and meets at least quarterly and provides a forum to discuss trial progress with key members and the content of reports to, and responses from, the oversight committees.

## REGULATORY APPROVALS

Ethical approval for the trial protocol was received on 18 September 2018 from Wales Research Ethics Committee (REC) 3—reference 18/WA/0092. Regulatory approval from the MHRA was received on 26 June 2018. Site-specific capability and capacity will be sought for the trial. Amendments to REC-approved documentation will not used until approval from the relevant regulatory authorities is in place.

**Author affiliations**
[1]Division of Population Medicine, Cardiff University, Cardiff, UK
[2]Swansea Trials Unit (STU), Swansea University, Swansea, UK
[3]Diabetes Research Unit Cymru (DRUC), Swansea University, Swansea, UK
[4]Diabetes Research Group, Cardiff University, Cardiff, UK
[5]Peter Gorer Department of Immunobiology, King's College London, London, UK
[6]National Institute for Health Research (NIHR) Biomedical Research Center (BRC), Guy's and St Thomas' NHS Foundation Trust and King's College London, London, UK
[7]Department of Molecular Genetics, Cumming School of Medicine, The University of Calgary, Calgary, Alberta, Canada
[8]Patient and Public Representative, Ammanford, UK
[9]Patient and Public Representative, Cardiff, UK

**Contributors** Authors MAA, NB, JB-M, AB, WYC, CD, GD, GWF, JWG, SH, GH, HAH, SL, SM-J, AM, RS, DT, TT, KC and JHMY played a significant role in the development of the protocol. Authors SM-J and AB are our PPI representatives and review the protocol and other trial related documentation. CD is the chief investigator whilst JG is the paediatric T1D lead and DT is the adult T1D lead. CD and DT are the joint senior authors of the paper.

**Funding** This project (project reference 16/36/01) is funded by the Efficacy and Mechanism Evaluation (EME) Programme, an MRC and NIHR partnership. Additional funding for mechanistic laboratory tests has been provided by JDRF (Juvenile Diabetes Research Foundation) International Award 3-SRA-2018–629 s-B.

**Disclaimer** The views expressed in this publication are those of the author(s) and not necessarily those of the MRC, NIHR or the Department of Health and Social Care.

**Competing interests** None declared.

**Patient consent for publication** Not applicable.

**Provenance and peer review** Not commissioned; externally peer reviewed.

**ORCID iDs**
John W Gregory http://orcid.org/0000-0001-5189-3812
Kymberley Carter http://orcid.org/0000-0003-0691-6282
Wai Yee Cheung http://orcid.org/0000-0002-0915-9312
Gail Holland http://orcid.org/0000-0002-6924-2521
Stephen Luzio http://orcid.org/0000-0002-7206-6530
Gareth Dunseath http://orcid.org/0000-0001-6022-862X
Timothy Tree http://orcid.org/0000-0002-6973-5377
Jennie Hsiu Mien Yang http://orcid.org/0000-0001-6171-833X
Ashish Marwaha http://orcid.org/0000-0003-1234-0224
Nadim Bashir http://orcid.org/0000-0003-0501-1342
Hayley Anne Hutchings http://orcid.org/0000-0003-4155-1741
Greg W Fegan http://orcid.org/0000-0002-2663-2765
Stephen Hiles http://orcid.org/0000-0002-8376-4377
Colin Dayan http://orcid.org/0000-0002-6557-3462

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
