## [Reviewer comments · BMJ Open]

ARTICLE DETAILS

TITLE (PROVISIONAL)	Phase II multi-centre, double-blind, randomised trial of ustekinumab in adolescents with new-onset Type 1 Diabetes (USTEK1D): Trial Protocol
AUTHORS	Gregory, John; Carter, Kymberley; Cheung, Ivy; Holland, Gail; Bowen-Morris, Jane; Luzio, Stephen; Dunseath, Gareth; Tree, Timothy; Yang, Jennie; Marwaha, Ashish; Ali, Mohammad; Bashir, Nadim; Hutchings, Hayley; Fegan, Greg; Stenson, Rachel; Hiles, Stephen; Marques-Jones, Susie; Brown, Amy; Tatovic, Danijela; Dayan, Colin

VERSION 1 – REVIEW

REVIEWER	Ziegler, Anette Institute of Diabetes Research, Helmholtz Zentrum München
REVIEW RETURNED	21-Apr-2021

GENERAL COMMENTS	The study protocol manuscript is very clearly presented. It is just a bit surprising that the study is still ongoing. The study started in 2018 (3 years ago) and has a planned recruitment period of 2 years. I suggest that the authors provide the exact PPFV (first patient first visit date), the number of adolescents already enrolled, and the expected LPLV.
---

REVIEWER	Ludvigsson, Johnny Linkopings universitet, Div of Pediatrics, Dept of Biomedical and Clinical Sciences
REVIEW RETURNED	22-May-2021

GENERAL COMMENTS	Congratulations to an important study. I think this drug should be studied at the onset of T1D (I was actually involved in designing a protocol for Stelara several years ago, but for some reasons the study was never done). I understand that your study is already ongoing, and then it is too late to make any revision except for some minor things I mention below. Even if the protocol can be accepted as it is , I have some concerns: - the study should have been double-blind for at least 18 months, best 24 months (even with that sort och clinical follow-up you have after 12 months). With the present design you can only draw conclusion on efficacy for a short time after the last active treatment is given.- I would have recommended fewer sites that 16 for such a small number of patients (72). With only 3-5 patients per site you introduce unnecessary large clinical variation, especially as your clinical recommendations are very vague. Only NICE guidelines, but nothing
---

	clear about blood glucose targets before and after meals etc - As exclusion you have: Any live immunisations for 1 month prior to trial entry., but nothing is said about immunisation during the trial!? Is live immunisations allowed during the trial?? If not Amendment should be done. Regarding methods I think you need to clarify: You have as secondary endpoint Exogenous insulin requirement as reflected in Number of responders (defined as participant who has HbA1c \leq 48mmol/mol and mean daily insulin use <0.5 IU/kg/day) measured over 7 consecutive days during the 2 weeks preceding the visit , but you do not say which these 7 days are out of the 14: Are the investigators allowed to chose the best consecutive 7 days?? Another unclear point: When MMTT is to be done and blood glucose >11.1 the patient is allowed to take correction dose of insulin, but it is not said how long time to wait until MMTT could be done? Then it is said that if blood glucose is still after two hours not below 11.1 the MMTT MAY be postponed. Thus, is it allowed to wait 3 hours? . This procedure with MMTT posible eg 30 minutes after correction dose of insulin is different from the recommendation for DBS when the patient is recommended to wait for 60 min before insulin is taken. A minor thing: Primary endpoint your write: Primary outcome: MMTT C-peptide AUC values at week 52 at week 52 (one "at week 52" can be removed).- Then I am a bit surprised that you do not have change in C-peptide from baseline to 12 months , but only values at 12 months. OK, but the decline would have been even more interesting if you want to study the efficacy of ustekinumab. What happens if the baseline C-peptide values are different between the groups? Finally I am just a bit surprised in the Information letters to individuals aged 16-18 years that nothing is said about adverse events and risks of ustekinumab. In our country that would hot have passed Research Ethics Board. And one can question for 12 years old children to give so much money: You will receive a £10.00 gift voucher for each treatment visit and we will give you £30.00 gift if you come to the final visit (visit 8) (that's £100 in total if you come to all visits)- But OK, I suppose that is OK.
--	---

REVIEWER	Dejgaard, Thomas Steno Diabetes Center Copenhagen
REVIEW RETURNED	16-Jul-2021

GENERAL COMMENTS	This very well written manuscript by Gregory et al. describes a protocol for a multi-center RCT evaluates ustekinumab vs. placebo treatment for 52 weeks on endogenous insulin production in adolescents. Overall, a very important and interesting study. The design and method used are clearly described and considered the golden standard for these kinds of investigations. The topic of this manuscript is found to be of interest for the reader of BMJ Open. I only have minor comments/suggestions to consider before publishing. Introduction P.3 ln.16: I think it would be relevant to comment on previous trials conducted with immunotherapy in T1DM eg. Teplizumab, Rituximab and Golimumab.
---

	Objectives P.4 In.26: You state that you measure glycemic variability, but mention only TIR and hypoglycemia, which is not describing glycemic variability? I suggest adding SD and CV. Consent A big applause for the youtube recruitment video Eligibility criteria P.6 In.17: What was the rationale to choose up to 100 days between diagnosis and start of treatment? It seems a quite long period with the risk of loosing even more endogenous production. MMTT P.9 In.3: Did you check for hypoglycemia in 24-48 hours before the MMTT? This could have impact on the response of glucagon and C-peptide. Long term follow-up P.10 In.32: Why not consider a MMTT for long term follow up? At least for a subgroup? I think it would be very interesting to see the long-term effect after stopping treatment? I know it isn't a must for a protocol publication, but a discussion about the considerations you have done in relation to trial design, group of patients etc. would have been valuable for the reader. I'm looking very much forward to see the results from this trial.
--	--

VERSION 1 – AUTHOR RESPONSE

Reviewer: 1

Dr. Anette Ziegler, Institute of Diabetes Research, Helmholtz Zentrum München Comments to the Author:

The study protocol manuscript is very clearly presented.

It is just a bit surprising that the study is still ongoing. The study started in 2018 (3 years ago) and has a planned recruitment period of 2 years. I suggest that the authors provide the exact FPFV (first patient first visit date), the number of adolescents already enrolled, and the expected LPLV.

RESPONSE: We thank the reviewer for her comments. The trial is still ongoing because of a pause to recruitment due to the COVID-19 pandemic and a delay in some participating sites reopening to recruitment thereafter. This has been clarified in the 'strengths and limitations' section. We are expecting to randomise our final participant this month (August 2021).

Reviewer: 2

Prof. Johnny Ludvigsson, Linkopings universitet Comments to the Author:

Congratulations to an important study. I think this drug should be studied at the onset of T1D (I was actually involved in designing a protocol for Stelara several years ago, but for some reasons the study was never done). I understand that your study is already ongoing, and then it is too late to make any revision except for some minor things I mention below.

Even if the protocol can be accepted as it is , I have some concerns:

- the study should have been double-blind for at least 18 months, best 24 months (even with that sort och clinical follow-up you have after 12 months). With the present design you can only draw conclusion on efficacy for a short time after the last active treatment .
is given.

RESPONSE: We thank the reviewer for his comments. Both participants and the research team remain blinded to the end of the trial, during which secondary outcome data including capillary

blood spot c-peptide profiles will continue to be collected, providing longer term data on the effectiveness of the intervention up to 24 months as suggested. We have clarified this on p12 of our manuscript.

- I would have recommended fewer sites than 16 for such a small number of patients (72). With only 3-5 patients per site you introduce unnecessary large clinical variation, especially as your clinical recommendations are very vague. Only NICE guidelines, but nothing clear about blood glucose targets before and after meals etc

RESPONSE: We chose 16 sites because of potential low recruitment rates per site, anticipated from the experiences in similar trials, of the Chief Investigator and other senior co-investigators. It is too late now to adjust the structure of the trial as suggested. However, if we demonstrate a difference in outcomes between the 2 arms of the trial, the larger number of sites used will suggest greater clinical utility for the intervention in routine clinical practice.

- As exclusion you have: Any live immunisations for 1 month prior to trial entry., but nothing is said about immunisation during the trial!? Is live immunisations allowed during the trial?? If not Amendment should be done.

RESPONSE: We apologise for an error in the paper. The trial protocol does state "Planned live immunisations are also not permitted during the study period" but this has been omitted from the table. We have added this to Table 2.

Regarding methods I think you need to clarify:

You have as secondary endpoint

Exogenous insulin requirement as reflected in Number of responders (defined as participant who has HbA1c \leq 48mmol/mol and mean daily insulin use $<$ 0.5 IU/kg/day) measured over 7 consecutive days during the 2 weeks preceding the visit, but you do not say which these 7 days are out of the 14: Are the investigators allowed to choose the best consecutive 7 days??

RESPONSE: We thank the reviewer for highlighting this. We have the details of our planned analysis in a statistical analysis plan which is separate to the protocol. We can confirm that we plan to take the 7 full days closest to the visit date for analysis, regardless of any variations in a longer term data-set.

Another unclear point: When MMTT is to be done and blood glucose $>$ 11.1 the patient is allowed to take correction dose of insulin, but it is not said how long time to wait until MMTT could be done? Then it is said that if blood glucose is still after two hours not below 11.1 the MMTT MAY be postponed. Thus, is it allowed to wait 3 hours? . This procedure with MMTT possible eg 30 minutes after correction dose of insulin is different from the recommendation for DBS when the patient is recommended to wait for 60 min before insulin is taken.

RESPONSE: We apologise for the lack of clarity on the point raised by the reviewer. Blood glucose levels prior to the MMTT were to be checked at home on waking. In practice therefore, it was planned that there should be a two hour gap between any subsequent insulin correction and undertaking the MMTT, after arrival at the clinical research facility. This has been clarified in the manuscript on p9.

A minor thing: Primary endpoint you write: Primary outcome: RESPONSE: We apologise but we are unclear what is being requested of us here, as the term 'primary outcome' is commonly used in trials such as this.

MMTT C-peptide AUC values at week 52 at week 52 (one "at week 52" can be removed). RESPONSE: We will amend this as requested.

Then I am a bit surprised that you do not have change in C-peptide from baseline to 12 months, but only values at 12 months. OK, but the decline would have been even more interesting if you want to study the efficacy of ustekinumab. What happens if the baseline C-peptide values are different between the groups?

RESPONSE: We can clarify that the values at 12m for each patient will be adjusted by their baseline data as stated in the data analysis section of the manuscript. In addition to adjusting for any baseline imbalance, this approach has the advantage of avoiding "regression to the mean" and provide a more stable estimate of the treatment effect.

Finally I am just a bit surprised in the Information letters to individuals aged 16-18 years that nothing is said about adverse events and risks of ustekinumab. In our country that would not have passed

Research Ethics Board. And one can question for 12 years old children to give so much money: You will receive a £10.00 gift voucher for each treatment visit and we will give you £30.00 gift if you come to the final visit (visit 8) (that's £100 in total if you come to all visits)- But OK, I suppose that is OK.

RESPONSE: We included reference to the mostly hypothetical risks we were aware of, based on treatments using monoclonal antibodies in general in this population. We liaised closely with the drug manufacturer when developing the information sheet and the safety information was reviewed by the UK Research Ethics Committee (REC) and

Medicines and Healthcare Products Regulatory Agency (MHRA) for approval.

We chose to offer vouchers for the participants because of the demanding home-based data and sample collection requirements (diary, dried blood spot card and wearing of a glucose monitor sensor). Our Patient and Public Involvement (PPI) representative fully supported the suggestion and felt that the amounts were reasonable but not coercive. Our Research Ethics Committee deemed these to be a reasonable reward and provided approval.

Reviewer: 3

Dr. Thomas Dejgaard, Steno Diabetes Center Copenhagen Comments to the Author:

This very well written manuscript by Gregory et al. describes a protocol for a multi-center RCT evaluates ustekinumab vs. placebo treatment for 52 weeks on endogenous insulin production in adolescents. Overall, a very important and interesting study. The design and method used are clearly described and considered the golden standard for these kinds of investigations. The topic of this manuscript is found to be of interest for the reader of BMJ Open. I only have minor comments/suggestions to consider before publishing.

Introduction

P.3 In.16: I think it would be relevant to comment on previous trials conducted with immunotherapy in T1DM eg. Teplizumab, Rituximab and Golimumab.

RESPONSE: We thank the reviewer for his comments. In the Introduction, we have now referred to results of clinical trials of immunotherapy (references 7-9).

Objectives

P.4 In.26: You state that you measure glycemic variability, but mention only TIR and hypoglycemia, which is not describing glycemic variability? I suggest adding SD and CV.

RESPONSE: We will include SD and CV as suggested by the reviewer in our statistical analysis plan.

Consent

A big applause for the youtube recruitment video

RESPONSE: We are pleased to see a positive response to our video and intend to advocate the use of short videos to introduce trials in paediatrics more widely.

Eligibility criteria

P.6 In.17: What was the rationale to choose up to 100 days between diagnosis and start of treatment? It seems a quite long period with the risk of losing even more endogenous production.

RESPONSE: We selected 100 days based on other T1D trials used by the Immune Tolerance Network and TrialNet previously. We felt that this time interval which would allow the beta cell metabolic response to stabilise following diagnosis and introduction of insulin treatment was the appropriate way to proceed. Additionally, it allowed us to expand the recruitment window so that patients who were diagnosed a number of weeks earlier still had an opportunity to take part.

MMTT

P.9 In.3: Did you check for hypoglycemia in 24-48 hours before the MMTT? This could have impact on the response of glucagon and C-peptide.

RESPONSE: This is a good point. We collect diary & continuous glucose monitor data on hypos prior to every study visit, not just the MMTT visits & will explore potential relationships between these documented episodes of hypoglycaemia and the c-peptide response to MMTT as suggested by the reviewer. However, screening MMTTs are usually done on the day of consent so we cannot retrospectively collect data about recent hypos at the screening visit though clinically significant hypos are relatively unusual so soon after diagnosis.

Long term follow-up

P.10 In.32: Why not consider a MMTT for long term follow up? At least for a subgroup? I think it would be very interesting to see the long-term effect after stopping treatment?

RESPONSE: We appreciate this suggestion which we agree with. It was originally part of the grant application for the trial but the funder was not prepared to support this level of follow up. We therefore opted instead to use the capillary blood spot c-peptide measures as a proxy for a formal MMTT c-peptide response.

I know it isn't a must for a protocol publication, but a discussion about the considerations you have done in relation to trial design, group of patients etc. would have been valuable for the reader. As presently drafted, our manuscript does not contain a Discussion section where such considerations could be reviewed. If the editor would like us to do so, then we will be happy to include such a section which likely could include discussion about the dose of ustekinumab we have used (influenced by previous clinical experiences in other disease states) and the age range selected (to compliment another trial of ustekinumab in adults in Canada).

VERSION 2 – REVIEW

REVIEWER	Ludvigsson, Johnny Linkopings universitet, Div of Pediatrics, Dept of Biomedical and Clinical Sciences
REVIEW RETURNED	26-Aug-2021
GENERAL COMMENTS	Thank you for adequate response to my questions and comments. Good luck with your trial.